# Numerical Analysis of Shear and Particle Crushing Characteristics in Ring Shear System Using the PFC^2D^

**DOI:** 10.3390/ma14010229

**Published:** 2021-01-05

**Authors:** Sueng-Won Jeong, Kabuyaya Kighuta, Dong-Eun Lee, Sung-Sik Park

**Affiliations:** 1Geologic Environment Division, Korea Institute of Geoscience and Mineral Resources, Daejeon 34132, Korea; swjeong@kigam.re.kr; 2Department of Civil Engineering, Kyungpook National University, 80 Daehakro, Bukgu, Daegu 41566, Korea; dankabuyaya@knu.ac.kr; 3Department of Architectural Engineering, Kyungpook National University, 80 Daehakro, Bukgu, Daegu 41566, Korea; dolee@knu.ac.kr

**Keywords:** residual shear stress, particle crushing, ring shear test, particle flow code (PFC^2D^), frictional work

## Abstract

The shear and particle crushing characteristics of the failure plane (or shear surface) in catastrophic mass movements are examined with a ring shear apparatus, which is generally employed owing to its suitability for large deformations. Based on results of previous experiments on waste materials from abandoned mine deposits, we employed a simple numerical model based on ring shear testing using the particle flow code (PFC^2D^). We examined drainage, normal stress, and shear velocity dependent shear characteristics of landslide materials. For shear velocities of 0.1 and 100 mm/s and normal stress (NS) of 25 kPa, the numerical results are in good agreement with those obtained from experimental results. The difference between the experimental and numerical results of the residual shear stress was approximately 0.4 kPa for NS equal to 25 kPa and 0.9 kPa for NS equal to 100 kPa for both drained and undrained condition. In addition, we examined particle crushing effect during shearing using the frictional work concept in PFC. We calculated the work done by friction at both peak and residual shear stresses, and then used the results as crushing criteria in the numerical analysis. The frictional work at peak and the residual shear stresses was ranged from 303 kPa·s to 2579 kPa·s for given drainage and normal stress conditions. These results showed that clump particles were partially crushed at peak shear stress, and further particle crushing with respect to the production of finer in shearing was recorded at residual shear stress at the shearing plane.

## 1. Introduction

Erosion and rainfall-induced mass movements could result in significant life loss and property damage in urban areas. After slope failure initiation, spreading mass movements are strongly related to the frictional characteristics of the movement stages. In particular, evaluation of residual shear stress is crucial when a significant propagation of mass movements is expected after the onset of slope failure. For the safety of ecosystems, rapid downward moving masses involving soil, rock, water or their combinations have been intensively studied [1,2,3,4,5]. Evaluating failure and post-failure processes of rapid landslides requires in-depth knowledge of various scientific disciplines, such as geomorphology, geomechanics, hydraulics, and rheology, to predict, prevent, and stabilize the mass movements. Moreover, shear stress is an important mechanical parameters necessary to understand the landslide mobilization. Various shear tests have been conducted to investigate shearing characteristics of geomaterials [6,7,8,9,10,11,12,13,14,15,16,17,18]. The shear strength of granular materials is still challenging to determine using both experimental and numerical methods.

Bagherzadeh-Khalkhali and Mirghasemi [6] have investigated the direct shear strength of coarse-grained soils using experimental and numerical analysis under different normal stresses. They found that the characteristics of coarse-grained soil varied from strain-hardening to softening at the shearing duration as the normal stress increased, and the internal friction angle decreased with stress level increase. Cabalar et al. [7] have performed triaxial and cyclic tests to assess the strength of different sands, considering the effects of the particle shapes. They found that crush stone sands extracted from the northern region of Cyprus show a significantly higher strength than sands obtained from Gaziantep because of their different shapes of sand particles. They have also claimed that the degrees of sphericity and roundness of the particles increase the strength of sand and can reduce its volumetric strain. The above-mentioned experimental tests exhibit limitations in evaluating the overall shear characteristics of diverse mass movements. The most extensively used shear tests are the direct shear, triaxial shear, and ring shear tests; each of these has advantages and limitations. In contrast to other shear tests, the ring shear test can measure shear stress for large displacement [18]. In addition, it is performed with advanced equipment capable of controlling the consolidation, drainage, and shearing speed under static and dynamic conditions. Numerical analysis is often used as a way to overcome the limitations of laboratory experiments.

Lobo-Guerrero and Vallejo [19] proposed a discrete element method to simulate the evolution of sugar particle crushing subjected to ring shear testing; they used the Particle Flow Code in two dimensions (PFC^2D^). The periodic movement of particles in the ring shear apparatus was simulated with two parallel periodic boundaries and two saw-toothed standard boundaries; these were made with several edges to model the shearing surface. They found that the residual coefficient of the sugar materials was maintained constant in spite of the particles being crushed. However, erroneous stress computations can occur when combining periodic space boundaries and standard boundaries because the periodic space is not compatible with the standard boundary in PFC [20]. Moreover, the roughened shearing surface with multiples edges negatively affects the particle response and the stress calculation, especially for particles distributed in the corners. In addition, neither the periodic space nor the standard boundary method take into account the interactions between machine components and particles.

A more accurate and efficient evaluation of the ring shear characteristics of highly mobile landslide materials using the PFC^2D^ is needed. To simulate the ring shear test, we used a general boundary, which is crucial and useful for simulating the interactions between granular materials and machine components. We modeled the particle crushing using the clump method and the frictional work technique from PFC^2D^. The numerical results validated the results obtained from the ring shear experiments. The shear stress was calculated considering four different shearing velocities (0.01, 0.1, 1, and 100 mm/s), drainage condition, and normal stresses varying from 20 to 150 kPa. The discrete element method in PFC^2D^ requires contact models involving micromechanical properties of the granular materials. Material mechanical properties obtained experimentally are taken as macromechanical properties and are computed using the micromechanical properties by trial-and-error in PFC^2D^ [20]. The obtained microproperties can be utilized to design the shear behaviors of the waste materials in landslide hazards using PFC.

## 2. Materials and Methods

### 2.1. Materials

The waste materials were sourced from Busan Metropolitan city, Korea. They are taken from Imgi mine deposits, where the landslide occurs due to intense rainfall [21,22,23]. The landslide materials are mainly contained sub-graded and angular grains composed of pyrite, kaolinite, sericite, pyrophyllite, and quartz. The sample used for laboratory shear ring testing was composed of 35% gravel, 63% sand, and 2% other fine materials (i.e., fine particles that are more than 50% of soil passes 0.075 mm sieve). Thus, the soil sample can be considered as coarse-grained sediments. Porosity, the ratio of volume of voids to the total volume of the soil, is approximately 40%. These waste materials are categorized as gravelly sandy soils. Their mean diameter, effective grain size, and the uniformity coefficient are 1.5 mm, 0.3 mm, and 5 mm, respectively. The materials used are very similar to typical landslide materials encountered in Korea. Table 1 summarizes the geotechnical properties of the materials used. This work focuses more on the numerical analysis; more details about sample preparation and material properties can be found in [21].

### 2.2. Experimental Program

The ring shear test is suitable for investigating the shear characteristics of landslides because it offers several advantages and permits the measurement of shear at large displacement; it can also be used to investigate the mechanical characteristics of sliding surfaces due to large shear displacements [14]. We performed laboratory ring shear testing with a ring shear apparatus designed at the Korea Institute of Geoscience and Mineral Resources (KIGAM). This machine can quantitatively simulate the consolidation, drainage, and shear velocity in static and dynamic loading conditions. The outer and inner diameters and the height of the shear box of the KIGAM ring shear apparatus were 250, 110, and 75 mm, respectively. The shear box consists of an upper and a lower boxes, as shown in Figure 1. During the ring shear test, the upper box is fixed and the lower one rotates. The shear surface is clearly visible after testing (Figure 1b). Landslides may occur in a diverse shape and size. Normal stress can be considered based on the soil thickness where the shear surface observes. Shear velocity is important in determining the shear strength with respect to the landslide movement rate. There are numerous types of landslides, which are ranged from very slow to very rapid speed. Drainage is one of important conditions in the landslide initiation, because it is directly related to the generation of pore water pressure in shear surface (or landslide movement). Drained condition can be applied for no pore water pressure condition; thus, it can be used to reproduce very slow landslides, such as a creep motion of clay-rich landslides (e.g., a few centimeter per year). In the experimental program, the valve located in the ring shear box is open; thus, the water can freely move during shearing. No pore water pressure occurs. However, undrained condition is specifically used for a relatively rapid landslide occurrence (e.g., higher than 1.8 m/hr). The same boundary conditions are used in the numerical analysis, as detailed in the next section. We experimentally measured the normal stresses, vertical displacement from a linear variable differential transformer, pore pressure, and torques. The parameters considered in the experimental tests are: normal stress, drainage condition, and shear velocity, as presented in Table 2. Details on the laboratory experiments are found in [21].

## 3. Numerical Model

### 3.1. Discrete Element Method Description in PFC^2D^ and the Clump Method

The particle flow code (PFC^2D^) developed by Cundall is a discrete-element-method (DEM)-based software designed to simulate the movement and interaction of stressed granular assemblies. Cundall and his colleagues [24,25,26] are among the frontier researchers to apply the discrete element method to the movement of granular assemblies. The granular assembly consists of different particles that displace independent of one another, and the interaction between particles occurs only at contact points or interfaces. The PFC assumptions are as follows: the particles are considered rigid bodies; a soft-contact approach characterizes the particles at contact points, where they are permitted to overlap; the contact between particles can be defined by bonds; the shape of particles is either circular or spherical, with unit thickness; the overlap magnitude is related to the force of contact by the force–displacement law [20]; the overlap is small compared to the particle size. It denotes the relative contact displacement in the normal direction. The overlap equation is given by:(1)$$U^{n}=\;\left\{\begin{array}{c} R^{\left[A\right]}+R^{\left[B\right]}-d\;\;\left(\mathrm{particle}-\mathrm{particle}\;\mathrm{contact}\right)\;\;\\ R^{\left[b\right]}-d\;\;\;\;\;\;\;\;\left(\mathrm{patricle}-\mathrm{boundary}\;\mathrm{contact}\right) \end{array}\right.$$
where $R^{\left[A\right]}\;$ and $R^{\left[B\right]}$ are the radii of particles in contact. $R^{\left[b\right]}$ and *d* are the radius of a particle in contact with a wall (boundary) and the distance between particles centers, respectively.

In addition, the calculation process involves applying alternatively a force–displacement formulation at contact points and the Newton’s second law to the rigid bodies. Thus, the motion of each rigid body due to contact and forces applied on it is determined by Newton’s second law; further, the update of contact forces computed from the relative motion at each contact is governed by the force–displacement law. This law is applied for both particle–particle and particle–wall (i.e., model boundary) contacts. The computational scheme is a time stepping algorithm that consists of applying repeatedly the law of motion to each particle, a force-displacement law to each contact, and a constant updating of wall positions as shown in Figure 2. A detailed description of the DEM in PFC can be found in [27].

In the past decades, the particles generated for any granular assembly were simply circular or spherical; however, with the modern DEM, one can create a general particle shape using two or more circular or spherical particles [28,29,30]. The process of creating a particle of any shape is termed the clump or clustering method [27]. The particle created in the granular assembly may be a two-, three-, or four-particle clump, depending on the number of the particles forming it. For example, to simulate a granular assembly containing triangular or square particles or grains of a more natural shape, one only needs to combine few predefined simple particles to create the intended particle shape. The creation of triangular or square-shaped particles is illustrated in Figure 3a,b. The contact model is defined only between clump particles. The contact stiffness model, slip and separation model, and bonding model are the three different contact models provided by the PFC. The bonding models are classified into two types: contact bond models can simply produce a force and parallel bond models that can produce both a force and a moment. Herein, we employed the parallel bond as a cementation material between clump particles because it provides efficiently rotational movement of the particles in the granular system.

### 3.2. Simulation Procedure

As it is extremely difficult to measure micromechanical properties of soil and rock materials in laboratory experimentation, in which only the macromechanical properties of materials can be measured, the micromechanical properties of synthetic materials in PFC^2D^ can be used to obtain the macromechanical properties of granular materials by the trial-and-error method [20]. Although the DEM simulation cannot take into account as many particles as used in an experimental sample, it does guarantee a good approximation [20]. PFC^2D^ version 4.0 supports up to 100,000 particles for one granular assembly. In this study, to reduce the computational time, 6830 particles were used to simulate the ring shear test.

We performed ring shear simulation to investigate the normal stress and shear velocity effects on the shear stress. The clump method, as demonstrated in the particle flow code PFC^2D^, was used to generate the granular systems (Figure 4). The clump particles were created with five circular particles (Figure 4a) and random size distribution in the assembly. The particle cementation material that bonded the clump particles was set to parallel bond (Figure 4b,c). Since there is no fluid connection at the contact between touching particles in PFC^2D^, both drained and undrained conditions were simulated using the parallel bond between clump-particles and the calibration process may be used to obtain the macro-properties [20]. The micromechanical properties of the Lac du Bonnet material [20] were used to simulate the waste materials. Table 3 presents the materials properties of the clump particle assembly system. Different normal stresses were installed in the assembly using the initial stress installation procedure. After normal stress installation, a ring-shaped boundary was created using a general wall mechanism (Figure 4d). The top section of the ring shear box was assumed to simulate the 3D ring shear experiment. The ring shear box was rotated by applying to the outer boundary rotational velocities (i.e., 0.01, 0.1, 1, and 100 mm/s).

Our results showed that the shear stress increases when the normal stress and shear velocity increase. The shear angular velocity creates a centrifugal force on the particles. The materials are tested under normal stress of 20, 40, 60, 80, 100, and 150 kPa. The material was generated in the ring-shaped vessel in order to produce an isotropic and well-connected granular assembly at a specified normal stress. To perform an accurate ring shear simulation, the clump particle assembly was created using with the material genesis procedure [20].

## 4. Results and Discussions

We employed a series of numerical models to investigate the effects of the normal stress and shear velocity on the shear stress. The microproperties designed in PFC enable the simulation of the macromechanical properties obtained from laboratory experiments using a trial-and-error procedure. The three main macroproperties considered in this study are the modulus of elasticity, peak stress, and residual stress. After the materials were generated according to the material genesis procedure, using the micromechanical properties listed in Table 3, we repeatedly conducted numerical ring shear tests. Then, the resulting numerical values were directly compared to the experimental results by matching macromechanical properties. To reproduce the relevant behaviors of the waste materials, we determined the appropriate microproperties by a calibration process in which the response of the synthetic material is compared directly with the measured response of the waste materials. The results obtained experimentally were compared to those obtained by numerical simulations. Based on the research findings, the followings are highlighted: (a) shearing time, (b) shear velocity, (c) normal stress, and (d) crushing phenomenon during shearing.

### 4.1. Shear Stress and Shearing Time

We compared the experimental and numerical results for shear stress characteristics during a period of 300 s in the ring shear system for given drainage and normal stress conditions. To examine the effect of drainage and normal stress on the shear stress, we plotted shear stress vs. shearing time curves at a shear velocity of 0.1 mm/s; the normal stress was constant during each test (Figure 5). There is a clear peak value in shear stress–time relationships regardless of drainage condition. For a normal stress of 25 kPa, the experimental and numerical evaluations revealed a slope difference of 0.81 and 0.28 kPa/s in drained and undrained conditions, respectively; while for a normal stress of 100 kPa, the slope difference between the experimental and numerical curves was 0.56 and 0.39 kPa/s in drained and undrained condition, respectively. In addition, under the undrained condition, numerical and experimental results showed the similar peak stress values of 10.1 and 13.7 kPa at normal stress of 25 and 100 kPa, respectively; a peak stress difference of 0.3 kPa was obtained under the drained condition for both 25 and 100 kPa normal stresses. These results show that the experimental and numerical results for both drained and undrained condition are in good agreement.

A sudden drop of the shear stress appeared for both the experimental and numerical curves after the peak shear stress was found. Regardless of the normal stress level and drainage conditions, the materials evaluated here presented a strain-softening behavior (Figure 5). For a normal stress of 25 kPa, the calculated and experimental times at which the peak stress was reached differed by 1.1 s for the undrained condition, and 8.5 s for the drained condition. For a normal stress of 100 kPa, the calculated and measured time needed to reach the peak stress differed by 5.7 s for the undrained condition, and 0.9 s for the drained condition. These differences might be due to the difference in the time step scheme used for the calculation of the shear stress in PFC^2D^ [20].

Furthermore, Figure 5 illustrates the residual shear stress induced by the resistance of the clump particles after the drop in peak shear stress. This resistance is due to inter-particle friction and inter-locking effect between clump particles. Thus, numerical analysis is an efficient way to explain particle rearrangement with respect to the reduction in shear strength. Stabilization is reached for 150–300 s for both drained and undrained conditions with various normal stresses. The shape of the clump particles is also crucial to create some resistance after the drop in peak stress. We assumed that the residual shear stress was the shear stress measured during the stabilization period that followed the sudden drop in peak shear stress. In the granular assembly, a progressive clump particle crushing occurred after the drop in peak shear stress.

For a normal stress of 25 kPa, the difference in residual shear stress between the experimental and numerical analyses was 0.4 kPa for both the drained and undrained conditions (Figure 5a,b). For a normal stress of 100 kPa, the residual shear stress obtained experimentally and numerically differed by 0.7 kPa and 0.9 kPa for drained and undrained conditions, respectively (Figure 5c,d). As a result, the larger the normal stress, the larger the difference. These results show that the residual shear stress values we obtained experimentally and numerically were in good agreement.

### 4.2. Shear Stress and Shear Velocity

The effect of shear velocity on the shear stress is far more specific than those of drainage and normal stresses. We examined the shear characteristics of the waste materials as a function of shear velocity with respect to the peak and residual shear stress values. Figure 6 shows the influence of shear velocity on the peak and residual shear stress under different drainage and normal stresses. In general, the shear stress increased with an increase of shear velocity for all given conditions (Table 4). For a normal stress of 25 kPa under the drained condition, for shear velocities of 0.01, 0.1, 1, and 100 mm/s, the difference in peak shear stress between the experimental and numerical analyses was 0.1, 0.3, 0.1, and 0.7 kPa, respectively; the residual shear stresses differed by 1, 0.4, 0.5, and 1.7 kPa, respectively. For a normal stress of 100 kPa under the drained condition, at shear velocity of 0.01 mm/s, the experimental value of the peak shear stress was in a similar range compared with that obtained from numerical analysis; at shear velocities of 0.1, 1, and 100 mm/s, the difference in peak shear stress between the experimental and numerical evaluations was 0.3, 0.2, and 0.3 kPa, respectively. It seems that shear stress is not strongly affected by low shear speed (i.e., 0.01 mm/s) in ring shear apparatus used. Moreover, the residual shear stresses differed by 0.4, 0.7, 1, and 2.3 kPa at shear velocities of 0.01, 0.1, 1, and 100 mm/s, respectively (Figure 6a,b).

For a normal stress of 25 kPa under the undrained condition, at a shear velocity of 0.1 mm/s, the numerical peak shear stress was similar to that obtained experimentally; at the shear velocities of 0.01, 1, and 100 mm/s, the difference in peak shear stress values obtained experimentally and numerically was 0.2, 0.1, and 1.8 kPa, respectively; the residual shear stresses differed by 1.5, 0.4, 0.6, and 2.2 kPa, respectively. For a normal stress of 100 kPa under the undrained conditions, at shear velocities of 0.01, 0.1, and 1 mm/s, similar peak shear stresses were obtained by both numerical and experimental evaluations; at a shear velocity of 100 mm/s, the difference in peak shear stress between experiment and numerical analysis was 0.1 kPa. The residual shear stresses differed by 1.23, 0.9, 1.9 and 3 kPa at shear velocities of 0.01, 0.1, 1, and 100 mm/s, respectively (Figure 6c,d). These results show that the shear stress increases with a shear velocity increase. Similar results were obtained by Fukuoka et al. [31].

### 4.3. Shear Stress and Normal Stress

Figure 7 presents the influence of normal stress on shear stress under the drained and undrained conditions, obtained by both numerical and experimental analysis. The shearing velocity (0.1 mm/s) was employed to examine the influence of the normal stress on the shear stress. For a normal stress of 20, 40, 60, 80, 100, and 150 kPa under the drained condition, the values obtained for the difference in peak shear stress between the experimental and numerical analysis were 0.2, 0.1, 0.1, 0.1, 0.3, and 0.2 kPa, respectively. At normal stress of 80 kPa, the experimental residual shear stress was similar to that obtained by numerical analysis; for normal stress of 20, 40, 60, 100, and 150 kPa, the residual shear stresses differed by 0.8, 0.1, 0.1, 0.7, and 0.1 kPa, respectively.

Under the undrained condition of the soil, the difference in peak shear stress between the experimental and numerical analysis was 0.1, 0.1, 0.1, 0.4, 0.2, and 0.9 kPa for normal stresses of 20, 40, 60, 80, 100, and 150 kPa, respectively. For the normal stress of 20, 40, 60, 80, and 150 kPa, the difference between the experimental and numerical residual shear stress was 0.1, 0.6, 0.1, 0.7, and 0.1 kPa, respectively. These results show that the shear stress increases as the normal stress increases. Similar results were found by several other researchers [31]. These experimental and numerical results are in good agreement. In particular, the residual shear stresses obtained from experimental and numerical analysis are very similar one another; however, the peak shear stress under the drained condition is gradually increasing with normal stress and has almost three times higher than the counterpart. It may be due to the fact that there is more strong interaction between particles under the drained condition and results in high shear resistance. Under the undrained condition, water may pay an important role in the crushing and breakage process of granular material. It seems that the lubrication effect of fine particles may occur under the undrained condition. As previously mentioned, after the sudden and abrupt drop of the peak shear stress, the post-failure of the shear characteristics was examined using the residual shear stress. This phase is mainly characterized by the particle crushing mechanism, which is discussed in the next section.

### 4.4. Particle Crushing Characteristics

Obtaining information about the micromechanics of particle crushing in laboratory experiments is very difficult. However, this obstacle can be overcome by simulating the particle crushing using the discrete element method. The various shapes of particles may be created with two or more single particles using the clump logic; the generated particles are considered as rigid bodies. In the discrete element method, crushing of the granular material is defined as breakage of one or more particles off the clump particle. Thus, one or more criteria are required for implementing the particle crushing. The clump particle can be partially or completely broken when the crushing criteria are satisfied. For this purpose, a user-defined function using FISH language in PFC^2D^ was implemented.

Crushing mechanisms occur from two different mechanisms: abrasion and overstressing. Abrasion occurs through friction, when a particle rubs on another particle and is progressively abraded or broken. Overstressing occurs when a crack is generated in the clump particle, which is broken into two or more smaller particles as the crack enlarges [19,32,33]; this crack is created by excessive application of forces including compressive, tensile, and diametrical forces. In this work, we used the abrasion mechanism to simulate the crushing of clump particles.

Previous studies have simulated crushing of particles in a system based on a single particle crushing experimental data. In many situations, particles crushing in a granular system may often occur simultaneously. In this study, particle crushing was modeled using the energy dissipated by frictional sliding at contact points between particles. This energy is termed frictional work [20]. As previously mentioned, the residual shear stress was maintained constant owing to the frictional resistance between clump particles. The crushing of particles occurs when the required quantity of frictional work is produced between particles; this frictional work quantity was used to evaluate the particle crushing process in our granular assembly. Figure 8 presents the frictional work and development of shear zone in ring shear tests. For the laboratory experiment, the frictional energy was assumed to be the area under the shear stress-time relationship curves after reaching the peak shear stress (Figure 8a). A progressive development of shear zone is related to the reduction in shear stress in landslides. It can be illustrated in ring shear box (Figure 8b). Compared to the initial state of shearing, the shear zone is getting larger and larger during shearing. Shearing may create the finer particles in shear zone and result in the reduction in shear strength (i.e., strain softening behavior) during shearing. According to the previous research findings, large particles can be concentrated in the center of ring shear box, small particles can be accumulated mostly at the lower part of ring shear box due to the vertical movement occurred in shearing [2,15,31]. In addition, for fine-grained sediments, the shear surface is very thin (e.g., less than 1 mm thick), but for coarse-grained sediments, the shear surface is larger with shearing time [2,31,34].

For the DEM simulation, the crushing of clump particles was allowed until the required frictional energy was reached in the granular assembly system, depending on the shearing velocity. The frictional work, $W_{f}$, is computed as [27]:(2)$$W_{f}=\underset{N_{c}}{\sum }\left[\left(F_{i}^{s}\right){\left(\Delta D_{i}^{s}\right)}^{slip}\right]$$
where $N_{c}$, $F_{i}^{s}$ and ${\left(\Delta D_{i}^{s}\right)}^{slip}$. are the number of contacts, average shear force, and increment of the slip displacement, respectively, at the contact for the current time step. The increment of the slip displacement produced over a time step Δt is given by:(3)$$\Delta D_{i}^{s}=V_{i}^{s}\Delta t$$
where $V_{i}^{s}$ is the relative shear motion at contact, and is calculated as:(4)$$V^{s}=\;\left({\dot{x}}_{i}^{\left[\emptyset^{2}\right]}-{\dot{x}}_{i}^{\left[\emptyset^{1}\right]}\right)t_{i}-\omega_{3}^{\left[\emptyset^{2}\right]}\left|x_{k}^{\left[C\right]}-x_{k}^{\left[\emptyset^{2}\right]}\right|-\omega_{3}^{\left[\emptyset^{1}\right]}\left|x_{k}^{\left[C\right]}-x_{k}^{\left[\emptyset^{1}\right]}\right|$$
where ${\dot{x}}_{i}^{\left[\emptyset^{j}\right]}$ and $\omega_{3}^{\left[\emptyset^{j}\right]}$ are the translational and rotational velocity of the entity $\emptyset^{j}$, respectively. These are expressed as:(5)$$\left\{\emptyset^{1},\;\emptyset^{2}\right\}=\;\left\{\begin{array}{c} \left\{A,\;B\right\}\;\left(\mathrm{particle}\;-\;\mathrm{particle}\;\mathrm{contact}\right)\;\;\\ \left\{b,\;w\right\}\;\left(\mathrm{particle}\;-\;\mathrm{boundary}\;\mathrm{contact}\right) \end{array}\right.$$
and $t_{i}=\left\{-n_{2},\;n_{1}\right\}$ ($n_{1}\;$ and $n_{2}$ are the unit normal vectors).

Frictional energy is an important mechanical property of granular materials. In this study, the frictional work–shearing time relationship is examined. Figure 9 presents the variation in frictional work computed from the measured shear stress and time response at a constant velocity (i.e., 0.1 mm/s) for different drainage and normal stress conditions; D-NS25 and UD-NS25 denote the drained and undrained condition for normal stress of 25 kPa, respectively. The frictional work increases linearly with time. The frictional work obtained at peak shear stress was 303, 190, 220, and 155 kPa·s for D-NS100, UD-NS100, D-NS25, and UD-NS25, respectively. The frictional work at residual shear stress was 2579, 2418, 2280, and 1387 kPa·s for D-NS100, UD-NS100, D-NS25, and UD-NS25, respectively. Interestingly, it can be seen that the frictional energy is much more sensitive under the normal stress conditions than under the drainage conditions. Compared to the others, there is a large difference in the frictional energy at the end of testing (i.e., 300 s) for the normal stress of 25 kPa under the undrained condition. The total shearing time (i.e., 300 s) can be one of the limitations in this study. Nevertheless, it is considered to be sufficient to understand the shear and crushing characteristics of landslide materials with respect to the initiation of slope failure. The condition of normal stress of 25 kPa indicates a relatively shallow soil thickness. It has approximately 1 meter thick in the field. Water moves freely through soil matrix in the drained conditions; however, in the undrained condition, water captured in or surrounding shear zone due to water infiltration during or after heavy rainfall event may result in a sudden reduction in shear strength and cause high mobilization of landslide materials.

The clump particle crushing was permitted after the granular assembly systems reached the corresponding frictional work at peak and residual shear stresses. Figure 10 shows a progressive occurrence of clump particle crushing at peak and residual shear stress. The blue arrows in Figure 10 indicate where progressive crushing mechanism occurred in the ring boundary; the clump particle crushing occurs mainly on the shearing surface at the outer ring boundary. The results showed that the clump particles were partially crushed at peak shear stress, i.e., one particle has been separated from the original clump particle (Figure 10a). The shearing area with the peak value is approximately 5%–7% of the total (see Figure 5). More particle crushing occurred in the residual shear stress state (Figure 10b) due to substantial friction work, i.e., a progressive crushing continued occurring. For a given shear displacement (or time), the number of particles is approximately 10–16 at peak shear stress (Figure 10a), but it is approximately 15–19 at residual shear stress (Figure 10b) when we select a specific part and look closely in the upper and lower parts of ring shear box. The crushing or abrasion of the clump particle made of several particles at peak shear stress progressively abraded as the frictional work increased; the crushing occurred mainly in the vicinity of the shearing boundary because of not only the shear stress, but also to compressive stress and the diametrical and centrifugal forces created in the shear ring granular assembly system. The compressive stress is mainly due to the applied normal stress, whereas the diametrical and centrifugal forces are produced between clump particles by the translational and rotational velocities in the granular assembly system. Other researchers have also found that the particle crushing occurred principally on the shearing surface [35].

## 5. Conclusions

In this study, a simple ring shear numerical model was developed to investigate the shear and particle crushing characteristics of granular materials. The peak and residual shear stresses are strongly affected by variations of the normal stress, shear velocity, and drainage conditions. As expected, the shear stress increases with an increase in normal stress and shear velocity, regardless of the drainage conditions. In general, there is a very good agreement between experimental and numerical results. For both drained and undrained conditions, the shear stress reaches a peak value rapidly and then undergoes a sharp drop followed by a period of variations before stabilizing (i.e., a typical strain-softening behavior). This may be related to the occurrence of clump particle rearrangement and crushing during shearing. Further, the differences in peak and residual shear stresses are much larger under the drained condition rather than under the undrained condition. In other words, it can be expected that under low normal stress and undrained condition the soil can be mobilized much more easily than in the opposite case.

The particle crushing phenomenon in ring shear test is analyzed using DEM because it affects directly the shear stress. Using the frictional work concept in PFC^2D^, a new FISH language was implemented in PFC to simulate the clump particle crushing at both peak and residual shear stresses. As the friction work increases monotonically, the frictional work at residual shear stress is greater than that obtained at the peak shear stress. Therefore, the clump particles were partially crushed at peak shear stress, i.e., one particle has been separated from the original clump particle made of five particles; further, more crushing occurred (and was visualized) during the residual shear stress state owing to substantial frictional work. This explains the progressive crushing mechanism in the simulation of the ring shear test using DEM. The crushing (or abrasion) of the clump particle made of four particles at peak shear stress progressed as the frictional work increased. The clump particle crushing mainly occurred in the vicinity of the outer ring boundary due to limited shearing. In future studies, the shear and crushing characteristics over a long period of shearing time should be investigated through three-dimensional analysis.

## Figures and Tables

**Figure 1 materials-14-00229-f001:**
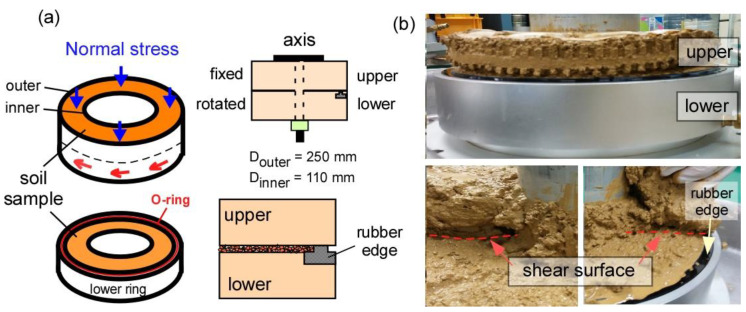
Schematical illustration of the ring shear box and shear surface after testing: (**a**) configuration of ring shear box and (**b**) observation of shear surface after testing.

**Figure 2 materials-14-00229-f002:**
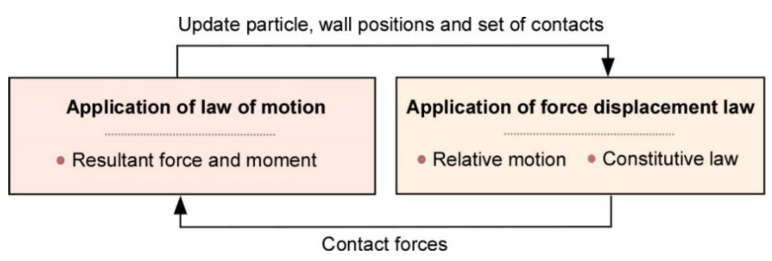
Discrete Element Method (DEM) computational scheme in PFC^2D.^

**Figure 3 materials-14-00229-f003:**
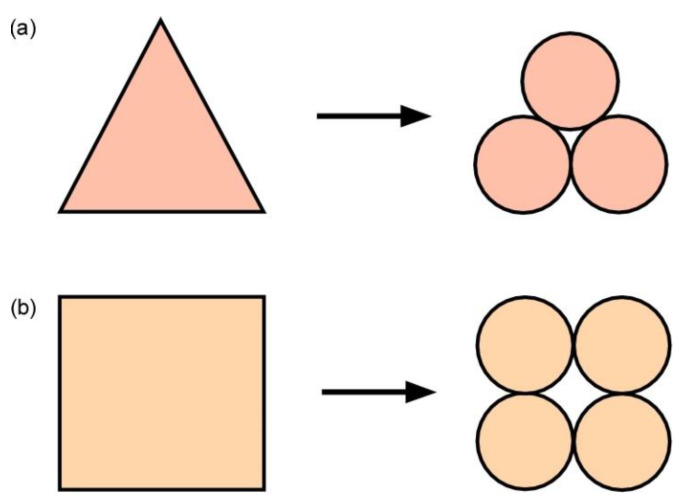
Illustration of clump formation: (**a**) Triangular particle made from nine overlapping circular particles and (**b**) Square particle made from 15 overlapping circular particles.

**Figure 4 materials-14-00229-f004:**
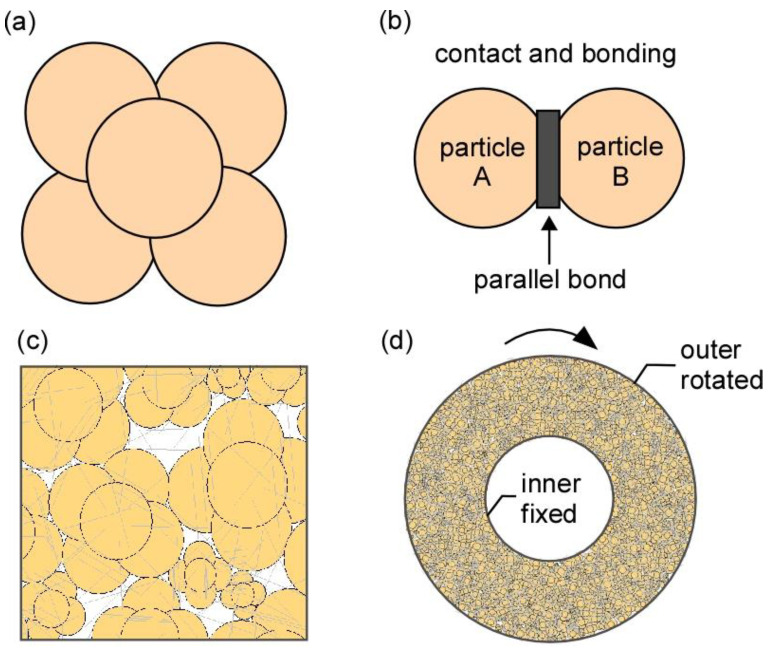
Clump particles and assembly: (**a**) clump particle, (**b**) parallel bond idealization between particles, (**c**) parallel bond connection between clump particles and (**d**) clump particles in ring shear box.

**Figure 5 materials-14-00229-f005:**
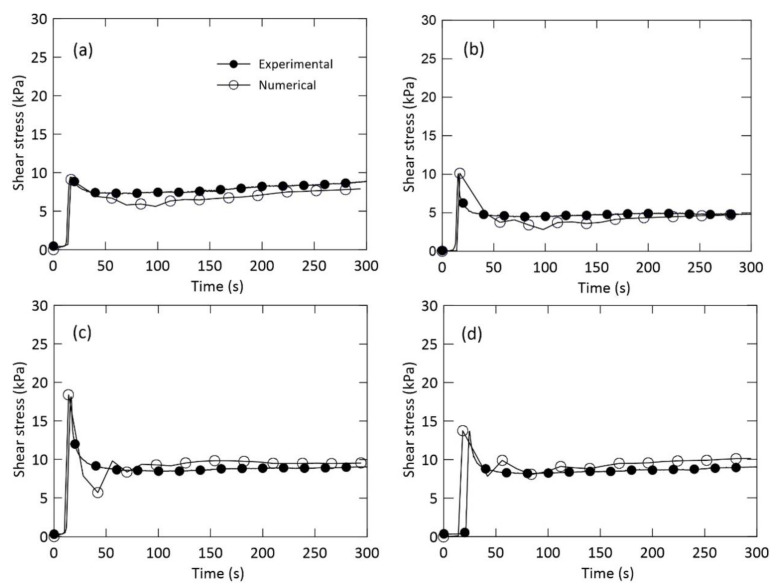
Shear stress-time curve in the ring shear system: (**a**,**b**) normal stress of 25 kPa and (**c**,**d**) normal stress of 100 kPa. (**a**,**c**) drained and (**b**,**d**) undrained condition at the same shear velocity of 0.1 mm/s.

**Figure 6 materials-14-00229-f006:**
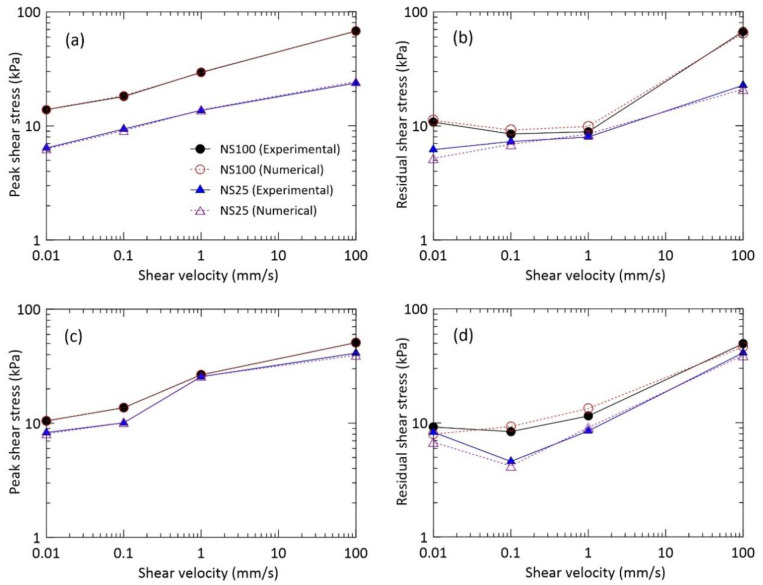
Peak and residual shear stress as a function of shear velocity: (**a**,**b**) drained and (**c**,**d**) undrained conditions.

**Figure 7 materials-14-00229-f007:**
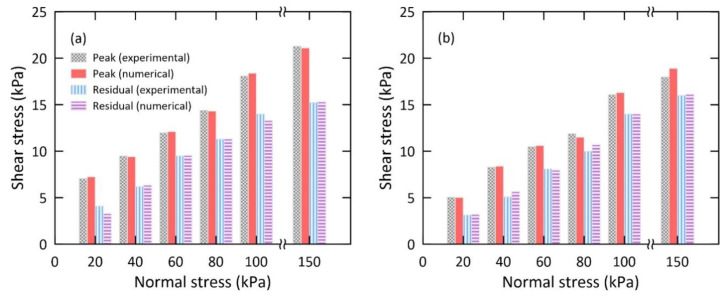
Shear stress vs. normal stress: (**a**) drained and (**b**) undrained condition.

**Figure 8 materials-14-00229-f008:**
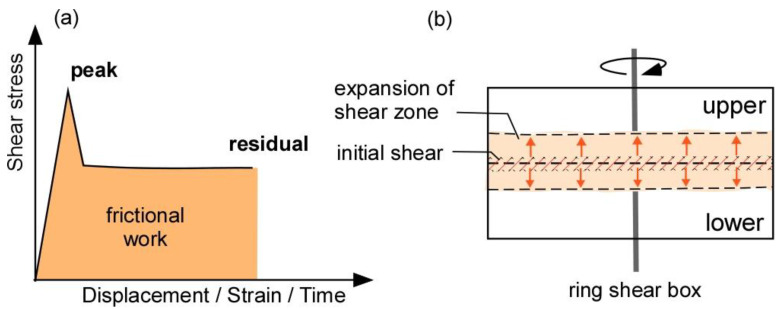
Frictional energy and shear zone in ring shear test: (**a**) shear stress-strain relationship to determine the peak and residual shear stress and (**b**) shear zone in ring shear box.

**Figure 9 materials-14-00229-f009:**
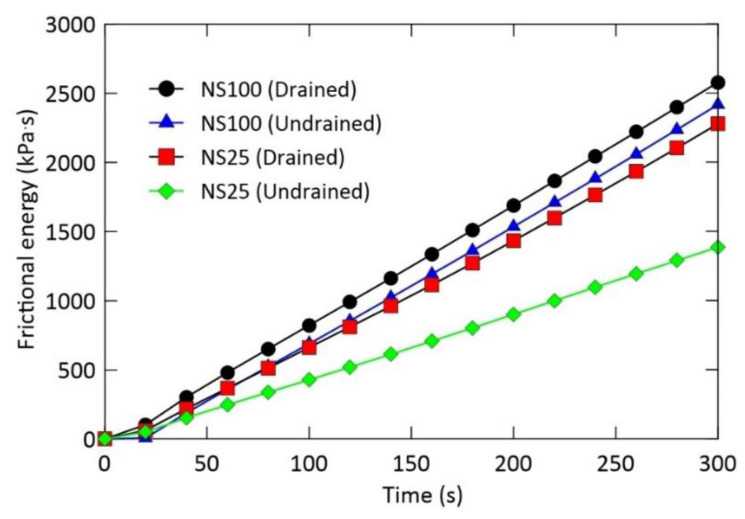
Frictional energy and shearing time dependent on drainage and normal stress condition.

**Figure 10 materials-14-00229-f010:**
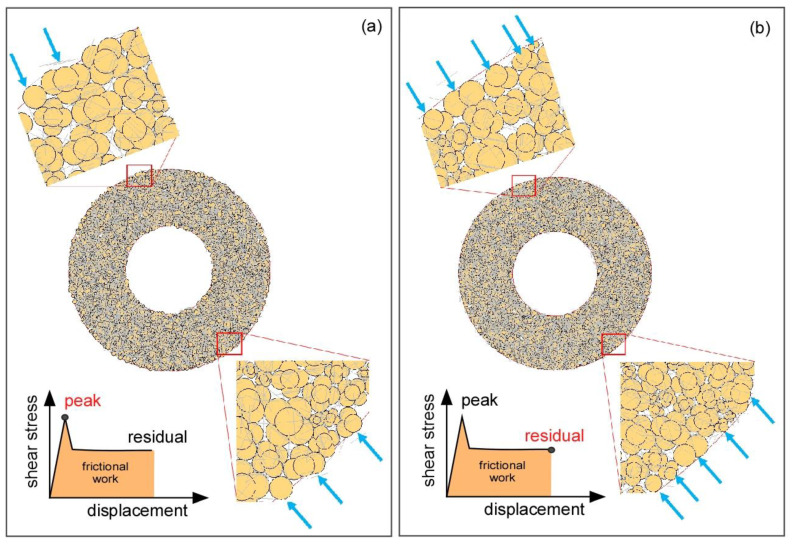
Clump particle crushing at: (**a**) peak shear stress and (**b**) residual shear stress. Arrows indicate a progressive occurrence of clump particle crushing.

**Table 1 materials-14-00229-t001:** Physical properties of waste materials.

Specific Gravity	Water Content (%)	Total Unit Weight (t/m^3^)	Dry Unit Weight (t/m^3^)	Liquid Limit (%)	Porosity (%)	USCS
2.63	6.9	1.7	1.59	24.5	39.5	SM

**Table 2 materials-14-00229-t002:** Experimental parameters.

Test Condition	Velocity (mm/s)	Normal Stress (kPa)
Drained	0.01 0.1 1 100	20
		40
		60
Undrained		80
		100
		150

**Table 3 materials-14-00229-t003:** Synthetic material properties.

Clump Particle	Cementing Material (Parallel Bond)
Bulk density 1700 kg/m^3^ R_max_/R_min_ ^1^ = 5.0 Modulus of elasticity = 6.1 MPa Normal to shear stiffness ratio = 2.5 Friction coefficient = 0.5	Bond-radius = 1 Modulus of elasticity = 6.1 MPa Normal to shear stiffness ratio = 2.5 Normal strength = Shear strength = mean ± std.dev = 162 ± 37 MPa

^1^ R_max_ = clump particle maximum radius, R_min_ = clump particle minimum radius, and std.dev = standard deviation.

**Table 4 materials-14-00229-t004:** Comparison of peak and residual shear stresses as a function of normal stress.

	NS25–τ_p_	NS25–τ_r_	NS100–τ_p_	NS100–τ_r_
Drained	τ = 12.8∙V^0.14^	τ = 10.3∙V^0.14^	τ = 29.5∙V^0.17^	τ = 17.3∙V^0.21^
Undrained	τ = 19.1∙V^0.18^	τ = 12.1∙V^0.21^	τ = 23.3∙V^0.17^	τ = 16.2∙V^0.19^

Note: NS = normal stress, τ_p_ = peak shear stress, τ_r_ = residual shear stress, and V = shear velocity (mm/s).

## Data Availability

Data sharing not applicable to this article as no datasets were generated or analysed during the current study.

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
