# Peer review of "Numerical Analysis of Shear and Particle Crushing Characteristics in Ring Shear System Using the PFC2D"

_materials, 2021, doi:10.3390/ma14010229_

Round 1

Reviewer 1 Report

Thank you for this contribution. This is an interesting and timely manuscript. This paper discusses how numerical analysis can be used to understand shear and particle crushing in ring shear systems using PFC2D. The conducted analysis is typically standard and falls within the expected work from such a publication and hence the work merits publication. As such, the authors are invited to properly address the following items:

  • The details on the model are quite superficial and light. Please ensure that your model is properly described to enable interested researchers from extending and replicating your work. Special attention should be paid to specifics such as element types, BCs, convergence criteria etc.

  • Please elaborate a bit on the contact between the particles. How was this contact modeled? 

  • How do particles clump together, numerically?
  • How does abrasion differs than crushing under shear?

Reviewer 2 Report

The manuscript can be very good but I have several comments and therefore I must suggest Major Revision based on the followings in attachment.

Reviewer 3 Report

The paper written by Sueng-Wong Jeong and her/his collaborators is potentially interesting but, as a referee, it is difficult for me to have definitive opinion on this paper due to a lack of clarity on the numerical methods. As a consequence, I can’t recommend publication of the manuscript in its present form.

At this stage of my understanding, my remarks are

  • About simulations, I don’t understand what the authors mean by « The displacement of different grain particles does not depend on the particles »
  • The authors wrote « the overlap should always be smaller compared to the particle size ». I would say « the overlap MUST always be MUCH smaller compared to the particle size . What is the typical value of overlap in the simulations?
  • What are the force laws used in this numerical work ? Please describe all of them precisely.
  • The authors wrote that « Modern DEM can create general particle shape using to or more circular or spherical particles ». This is true, but some works also deal with polyhedral particles. They deserve to be cited especially those dealing with particle crushing. For example (doi.org/1016/J.JMPS.2018.09.030 or doi.org/10.1140/epje/i2018-11656-1 among others)
  • What do the authors mean by random size generation? Does the size distribution used numerically is the same than the experimental one?
  • I do not understand, in the simulation which part of the system is rotated. I guess it is the outer circle. Please indicated it clearly on Fig 3d
  • Is the volume fraction homogenous within the shear cell?
  • The authors should report the radial velocity profile to shown how shear is localized as well as the temporal evolution of shear localization.
  • I don’t understand precisely the protocol used to build the initial state in the simulations.
  • I don’t understand how drainage is taken into account in the simulations. Please clarify
  • I don’t understand what the authors mean by « To evaluate the overall envelop of the curves obtained from ring shear experiments, the elastic modulus, peak stress and residual stress were successively adjusted by the trial-and-error method”
  • From figure 3a and the crushing mechanism, I understand that the latter does not conserve the volume of particles. The volume of a « clump particle » is indeed smaller than 5 times the volume of an individual particle. If I am right, this point is very problematic.
  • Figure 9, please indicate in the caption the meaning of the arrows.

Round 2

Reviewer 2 Report

The authors supplemented my comments and now the manuscript is correct. I recommend an article to publish in Materials.

Author Response

Thank you for your time and consideration.

Reviewer 3 Report

In their answer, the authors have addressed some of my comments in a convenient way. Yet, some of their responses remain very unclear. I listed them below. Most of them are minor, but comment #4 is important because masse conservation has to be verified!

  1. My comment about force-law was not understood and I appreciate that in their answer the authors try to learn me Newtonian mechanics. I meant what is the force-displacement law used. In the text the authors refer to the PFC manual (or something like that). I do not know at all PFC but I guess that several force-displacement laws are available. Which ones are chosen?
  2. I still do not understand how drainage is taken into account in the simulation. The authors’ response deals with experiments not simulations.
  3. The authors did not answer my question about the initial state. How the packing is initially built? Also the authors wrote “Then, the calibration was performed by trial-and error method by varying material property values to obtain the experimental mechanical properties (i.e. the curves obtained from the experiments). » I do not understand the material properties are supposed to be known in DEM simulation. Please clarify.
  4. I’m not at all convinced by the authors’ response on volume (and thus mass) conservation. In fact we should talk about surface since simulations are 2D. Look at figure 3. A triangular particle is made of 10 overlapping particles. Its surface is smaller than the total surface of the 10 individual particles (they overlap). So if this triangle particle crushes it surface increases (from the surface of the triangle to the surface of the 10 individual particles). If the surface is not constant, mass increases after crushing! In fact I didn’t understand
  5. Please indicate in the caption of figure 9 the meaning of the blue arrows!
